# GMR-Net: Road-Extraction Network Based on Fusion of Local and Global Information

**Zixuan Zhang [1], Xuan Sun [1,2,\*] and Yuxi Liu [3]**

1   Zhou Enlai School of Government, Nankai University, Tianjin 300350, China
2   Digital City Governance Laboratory, Nankai University, Tianjin 300350, China
3   College of Electronic Information and Optical Engineering, Nankai University, Tianjin 300350, China
*   Correspondence: sunxuan@nankai.edu.cn

**Abstract:** Road extraction from high-resolution remote-sensing images has high application values in various fields. However, such work is susceptible to the influence of the surrounding environment due to the diverse slenderness and complex connectivity of roads, leading to false judgment and omission during extraction. To solve this problem, a road-extraction network, the global attention multi-path dilated convolution gated refinement Network (GMR-Net), is proposed. The GMR-Net is facilitated by both local and global information. A residual module with an attention mechanism is first designed to obtain global and other aggregate information for each location's features. Then, a multi-path dilated convolution (MDC) approach is used to extract road features at different scales, i.e., to achieve multi-scale road feature extraction. Finally, gated refinement units (GR) are proposed to filter out ambiguous features for the gradual refinement of details. Multiple road-extraction methods are compared in this study using the Deep-Globe and Massachusetts datasets. Experiments on these two datasets demonstrate that the proposed method achieves F1-scores of 87.38 and 85.70%, respectively, outperforming other approaches on segmentation accuracy and generalization ability.

**Keywords:** road extraction; attention mechanism; multi-scale feature; local and global information

## 1. Introduction

With the development of remote sensing and computer science, high-resolution remote-sensing images are extensively applied in have received extensive attention in disaster management, urban planning, and other fields [1–3]. Obtaining target information quickly and intelligently from high-resolution remote-sensing images is an urgent challenge to be solved in the remote-sensing community today. Road-based geographic information serves city planning [4], vehicle navigation [5], geographic information management [6,7], etc., and is one of the key contents in the target extraction of remote-sensing-image extraction.

Road segmentation of remote-sensing images is a very challenging task [8] that essentially belongs to the classification of pixels. In such images, each pixel is classified and recognized. Compared with the general target segmentation, road segmentation is unique and complex. As important geographic information, roads are often affected by various factors [9], resulting in low segmentation accuracy. For example, (1) The narrow and connectivity of the road determines its small proportion in the whole image; (2) long and narrow roads will be blocked by vegetation, buildings and their shadows, making it more difficult to extract them from high-resolution remote sensing images; and (3) desert, bare soil, etc., have similar texture and spectral features with roads, which will also increase the difficulty of extraction.

Among the traditional road-extraction methods, Sun et al. [10] proposed a high-resolution remote-sensing-image road-extraction method based on the fast-progress and mean-shift methods. Road nodes are used as input and the mean-shift method is then used to initially divide them. Finally, the road between the set nodes is extracted by the fast-progress method. Anil et al. [11] proposed a method based on the active contour model.

This method uses a median filter to pre-process the image, enters the initial seed point, and then uses the active-contour model to extract the road. Chen et al. [12] established the global features of the road by automatically merging the road vector and road skeleton, and then extracted the local features of the image under the global feature constraints. It can be seen that the traditional methods mostly start from the morphological structure of the road, but with the improvement in remote-sensing image resolutions, complex situations reduce the effectiveness of such methods. With fast development in deep learning, increasingly more segmentation networks have been developed in recent years. As the first end-to-end learning network, fully convolutional networks (FCNs) [13] use convolution, up-sampling, and skip structures to achieve pixel-level classification. The target segmentation effect under complex conditions is poor due to the limited receptive field. Later, multi-scale [14] context semantic fusion modules were proposed, such as the Spatial Pyramid Pooling (SPP) module of the Pyramid Scene Parsing Network (PSPNet) [15] and the Atrous Spatial Pyramid Pooling (ASPP) module of deeplabv3 [16], which fully utilized context information [16]. Compared with traditional methods, neural networks can automatically extract multiple features other than colors, such as textures, shapes and lines. With the ability to automatically extract high-dimensional features, neural networks have been widely used in image fields, such as image classification, scene recognition, target detection, and semantic segmentation. Several scholars have also applied it to the field of remote sensing. Hong et al. provided a baseline solution for remote sensing image classification tasks using multimodal data by developing a multimodal deep learning framework [17]. Hong et al. proposed a mini graph neural network (miniGCN) that enables the combination of CNN and GCN for hyperspectral image classification [18]. Wang et al., proposed a new tensor low-rank and sparse representation method for hyperspectral anomaly detection [19]. Zhu et al., effectively extracted and fused global and local environmental information through an attention-enhanced multi-path network. The network uses multiparallel paths to learn multi-scale features of the space and attention modules to learn channel features for accurate extraction of building footprints and precise boundaries [20].

Some researchers apply neural networks to road extraction. For example, Chen et al. investigated the methods of automatic road extraction from remote sensing data and proposed a tree structure to analyze the progress of road extraction methods from different aspects [21]. Tamara et al. [22] proposed a road-segmentation model that combined a residual network with a U-Net network and used the residual structure to deepen the network to extract strong semantic information features. Zhang et al. [23] defined a DCGAN with specific conditions and achieved road segmentation by continuously optimizing the relationship between the generation network and confrontation network. Zhou et al. [24] improved the D-Linknet by adding a dilated convolutional layer based on LinkNet [25], using dilated convolution to expand the receptive area and retain spatial information, and fusing contextual information on multiple scales. Zhou et al. propose a new fusion network to fuse remote sensing images and location data to play the role of location data in road connectivity inference. A reinforced loss function is proposed to control the accuracy of road prediction output, which improves the accuracy of road extraction [26]. Yan et al. [27] proposed HsgNet based on global higher-order spatial information, modeled by bilinear pooling to obtain the feature distribution of weighted spatial information. Wan et al. [28] proposed a dual-attention road extraction network and constructed a new attention module to extract road-related features in spatial and channel dimensions, which can effectively solve the problem of road extraction discontinuity and maintain the integrity of roads. Li et al. [29] proposed a cascaded attention enhancement module considering multi-scale spatial details of roads to extract boundary-refined roads from remotely sensed images. Liu et al. [30] proposed a road extraction network based on channel and spatial attention (RSANet). Huo et al. [31] proposed a remote sensing image road extraction method with completion UNet, which introduces multi-scale dense dilation convolution to capture road regions.



The high-resolution remote-sensing image provides detailed road information but with significant noise [21]. In addition, the road structure is more complex. The global information of roads affects the structure and continuity of the road, and the local information affects the details of the road. Extracting and combining global and local information are very important for road segmentation. In the encoding-decoding network, U-Net [32], LinkNet, D-Linknet, and other networks only use simple convolution and pooling operations to extract features. The global information of the road is not fully taken into account, and no further attention is paid to the dependence between channels on the same level. Although the HsgNet method considers global information, it is indistinguishable from the above-mentioned network in combining global and local information, and are all connected by concat through a skip structure. Compared with deep features, shallow features have more location information, but their semantics are weaker and road features are not obvious [33]. The features directly supplemented by the skip structure have vague and ambiguous information, which is not conducive to refining the details. Therefore, this study aims to efficiently extract and fuse global and local context information to reduce the interference of fuzzy features [34,35], ensure the representativeness and usefulness of road features, and improve the accuracy of target segmentation. Finally, a high-resolution remote-sensing road-extraction network (GMR-Net) is proposed. The specific contributions of this study are the following.

- A new segmentation network for road extraction, called GMR-Net, is proposed, in which the encoding part uses the GC block attention module to enhance the focus on global information, and the decoding part filters out useless features by gating units to refine the segmentation details.
- To verify the accuracy and generalization ability of the model, experiments were conducted on the DeepGlobe Road Extraction dataset [36] and Massachusetts Roads dataset [37]. Experimental results show that, compared with D-Linknet, U-Net, RSANet, and PSPNet, the method proposed in this study achieves the expected results and shows better performance.

The rest of this article is organized as follows. In Section 2, the GMR-Net high-resolution remote-sensing road-extraction method is introduced. Experimental details and results are presented in Section 3 and discussion in Section 4. Conclusions are given in Section 5.

## 2. Method

This study proposes a method for extracting roads from high-resolution remote-sensing images. First, the original remote-sensing image is pre-processed. To prepare the training dataset, all the remote-sensing images and corresponding label images are intercepted by a fixed-size sliding window. Then, the designed deep neural network GMR-Net model is used to extract the road. All the pre-processed samples are used as the model's input, and the binary classification maps predicted by feature maps at different scales in the decoding stage are the output. The road-extraction result categories are "road" and "others".

### 2.1. Structure of the Deep Convolution Neural Network

The GMR-Net proposed in this article consists of three parts, as shown in Figure 1. The first part is the encoding end, consisting of a residual network with a global attention mechanism. The second part is multi-path dilated convolution (MDC), which extracts more comprehensive local and global information. The third part is the gated refinement unit (GR), which fully fuses the extracted local and global information. The refinement unit realizes the gradual refinement of the segmentation results, and the gating unit selects favorable features to supplement the details.

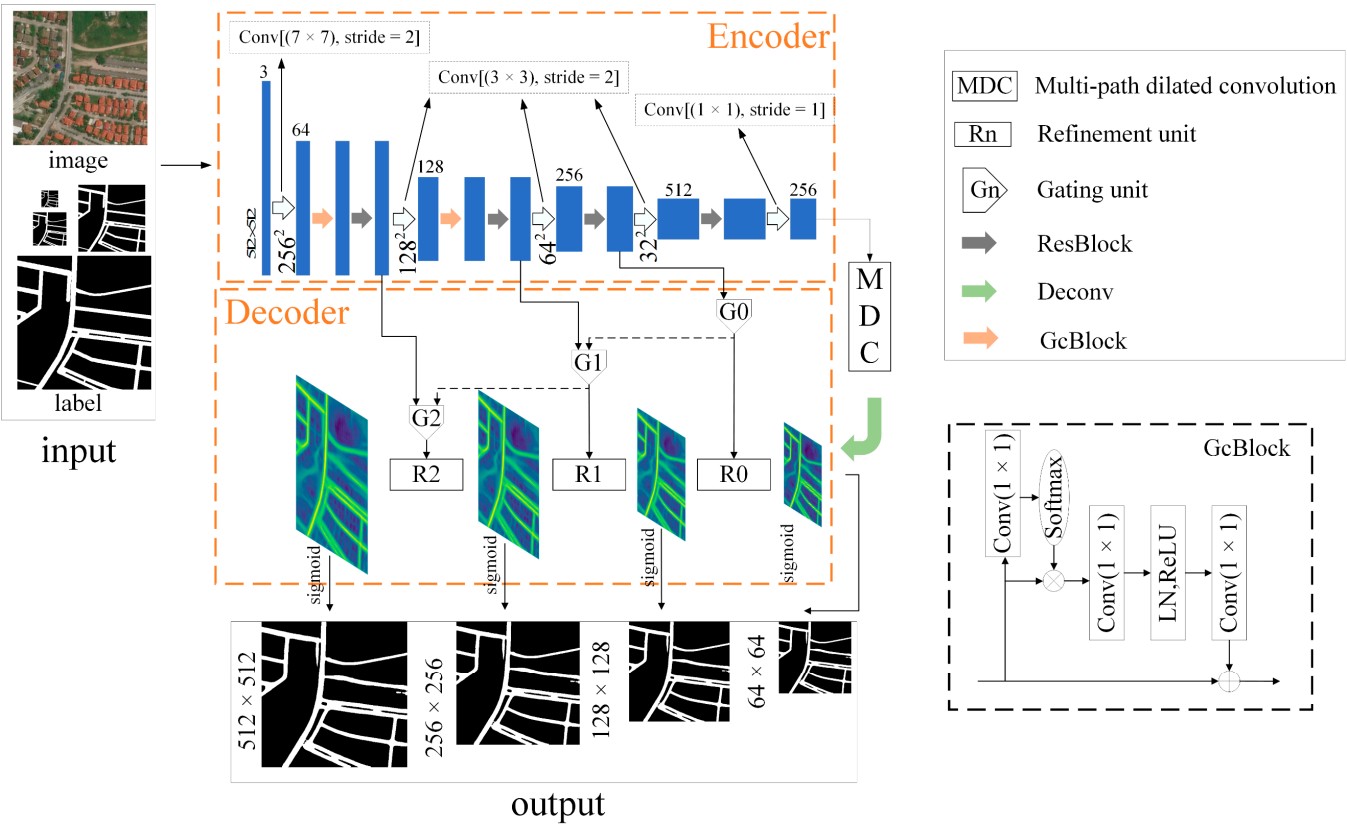

**Figure 1.** A schematic of the GMR-Net architecture.

Roads occupy a small proportion of the entire map due to the slenderness and connectivity of road structures. Therefore, it is particularly important to strengthen the focus on global context information. To better focus on extracting road features, the structure of ResNet34+GCblock is used to extract remote-sensing-image features. We use Resnet as the backbone network, and Resnet uses residual connections to solve the problem of disappearing gradients in the deep network. ResNet34 has a smaller number of parameters and can extract richer feature information of the image compared to ResNet18.

Among these, ResNet34 [38] has been pre-trained on the ImageNet dataset to accelerate the convergence rate of the model through transfer learning. Compared with the common residual network, GC-resnet adds the GC block global attention module [39] after each down-sampling. The GC-block module, a lightweight attention module formula, is expressed as (Equation (1)):

$$z_i = x_i + W_{v2}\text{Re}LU(LN(W_{v1}\delta)),$$
$$\delta = x_i \times Softmax(W_k x_i) \tag{1}$$

where $x = \{x_i\}_{i=1}^{N_p}$ is the input feature, $z = \{z_i\}_{i=1}^{N_p}$ is the output feature. $W_{v1}$ and $W_{v2}$ represent two $1 \times 1$ convolutions that change the number of channels of the feature map for capturing the relationship between channels. $\delta$ represents the attentional feature map generated after the global attentional pooling operation for the input features $x$. LN represents the normalized layer LayerNorm and ReLU represents the nonlinear activation function. $W_k$ represents $1 \times 1$ convolution.

The purpose of adding LayerNorm (LN) [40] is to reduce the difficulty of optimization and improve the generalization ability as a regularization. Finally, the output $z$ is obtained by feature fusion through broadcasting. Although GC-resnet deepens the number of network layers and extracts deep features, it pays more attention to global context information, focuses attention on the target road, and avoids learning background features.

The proposed GMR-Net is a single-input, multi-output learning network. An input remote-sensing image corresponds to four prediction output segmentation maps of different sizes.

After the front end extracts the features with global information, the MDC module is then used to obtain different sizes of perceptual fields. The context information is fully used through different degrees of receptive fields to extract road features of different size areas. Finally, in the decoding stage, the local and global context information is further efficiently aggregated using GR. Among these, the gating unit takes deep and shallow features as input and uses the deep semantic features to filter out the fuzzy information in the shallow features. The refinement unit is used to predict the segmentation results of each stage and gradually refine the details based on the prediction of the previous stage. Before each output result, the model uses a $1 \times 1$ convolution to map features to "road" and "others".

The training part uses backpropagation to calculate the gradient and update the network weights. Considering the fineness of roads, the road area accounts for a very small proportion of the entire high-resolution image. So, Dice loss, which is based on binary cross-entropy, is introduced [41]. Dice loss establishes the correct balance between the target foreground and background.

The loss of binary cross-entropy is defined as follows (Equation (2)):

$$loss_{Bce} = -\frac{1}{N} \sum_{1}^{N} [y_i \log(\hat{y}_i) + (1 - y_i) \log(1 - \hat{y}_i)] \tag{2}$$

where $N$ means the number of samples used in network training, $y_i$ means the label of the sample, and $\hat{y}_i$ means the result of the sample predicted to be road category after network prediction. The function of Dice loss is (Equation (3)):

$$loss_{Dice} = 1 - \frac{2\sum_{i}^{N} p_i g_i + smooth}{\sum_{i}^{N} p_i^2 + \sum_{i}^{N} g_i^2 + smooth} \tag{3}$$

where $N$ is the total number of pixels in each image, $g_i$ represents the ground-truth value of the $i$ pixel, and $P_i$ is the confidence score of the $i$ pixel in prediction results, and smoothing is set to 0.0.

To accelerate the convergence of the network and improve segmentation accuracy, predictions are made for the binary maps of each refinement stage [42]. Since every stage requires prediction, the nearest down-sampling is performed on the label to make its sizes $64 \times 64$, $128 \times 128$, $256 \times 256$, and $512 \times 512$. The final loss function is defined as (Equations (4) and (5)):

$$loss_{64}, loss_{128}, loss_{256}, loss_{512} = loss_{Bce} + loss_{Dice} \tag{4}$$

$$loss = loss_{64} + loss_{128} + loss_{256} + loss_{512} \tag{5}$$

where $loss_{Bce}$ is the binary cross-entropy function, i.e., Equation (2), and $loss_{Dice}$ is the Dice loss function, i.e., Equation (3).

## 2.2. Multi-Path Dilated Convolution

The road as a complex continuum, the surrounding trees and buildings, and its inflection points as local information often affect the accuracy of road segmentation [43]. Efficiently extracting local and global information is particularly important. To this end, the central part of the network is designed as an MDC that includes cascade and parallel modes. MDC combines dilated convolutions with different dilation rates to make the

combined receptive fields different. Features can extract more comprehensive local and global information through the attention of different receptive fields, as shown in Figure 2.

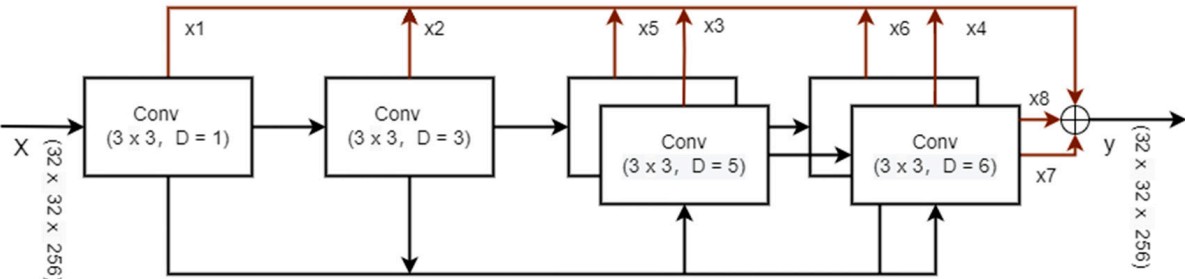

**Figure 2.** A schematic of the multi-path dilated convolution architecture.

Considering that the resolution of the input feature is $32 \times 32$, the dilation rate D in the design structure is 1, 3, 5, and 6. Through multi-path combination, this module can form eight scales, i.e., (1), (1, 3), (1, 5), (1, 6), (1, 3, 5), (1, 3, 6), (1, 5, 6), and (1, 3, 5, 6), and Equation (6) can be used to calculate the receptive fields, i.e., 3, 9, 13, 15, 19, 21, 25 and 31. In the end, the large receptive field covers the input features well, realizing multi-scale.

$$r_1 = d \times (k-1) + 1, r_n = d \times (k-1) + r_{n-1} \tag{6}$$

where $d$ is the dilation factor, $k$ is the size of the convolution kernel, and $r$ denotes the size of the convolution kernel after the ordinary convolution kernel is passed through the dilation design. $r_{n-1}$ denotes the size of convolution kernel before multi-layer dilation convolution, and $r_n$ denotes the size of convolution kernel after multi-layer dilation convolution.

When feature x is input, the feature channel is first compressed by a $1 \times 1$ convolution. Next, the compressed features are fed into the multi-path structure to obtain $x1$, $x2$, $x3$, $x4$, $x5$, $x6$, $x7$ and $x8$. To prevent the loss of feature-space information resulting from the discontinuity of the dilated convolution kernel, the design includes a convolution kernel with a dilation rate of 1 to fill the gap [42]. Finally, the features passing through different receptive fields are added to extract the context information of different scales before outputting the features. The specific definitions are as follows (Equation (7)):

$$\begin{aligned} x1 &= C_1(x) \\ x2 &= C_3(C_1(x)) \\ &\cdots \\ x8 &= C_6(C_5(C_3(C_1(x)))) \\ y &= x + x1 + x2 + \cdots x8 \end{aligned} \tag{7}$$

where $x$ represents the input, $y$ the output, and $C_n$ is the convolution with a dilation rate of $n$.

### 2.3. Gated Refinement Unit

As the network deepens, features have strong semantics, while shallow features have more location information [44]. To avoid the loss of detailed information in the encoding-decoding network, most networks use a skip structure to supplement the details in the decoding process. However, the number of convolutions corresponding to the coding layer is less, and the extracted local information and global information are not fully fused. Therefore, although the road features have more detailed location information, they do not have sufficient semantics. The supplementary information is mixed with fuzzy information. Here, the features of the corresponding coding layer and the deeper features are sent to the gating unit, and the deep semantic features are used to assist the shallow features in recovering the fuzzy information. After the gating unit outputs the features, it is processed into a binary map, and the refinement unit is used to gradually refine the

details of the binary map and separately predict the refined binary map. This process fully fuses the extracted local and global information and appears in every upsampling stage. By fusing the global and local information of roads, the fuzzy and ambiguous information brought by the shallow features is avoided, and the detailed information of roads is better supplemented. The detailed network structure is shown in Figure 3.

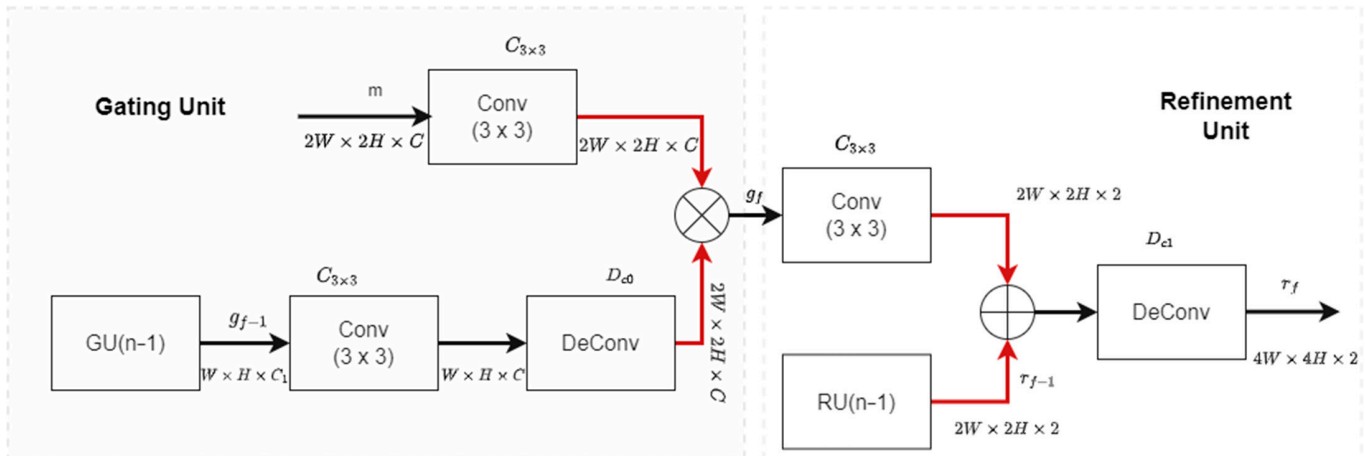

**Figure 3.** A schematic of the gated refinement unit architecture.

GR consists of two parts, the gating unit (GU) and the refinement unit (RU). The GU first takes the features $m$ in the corresponding coding layer and the features $g_{f-1}$ of the previous GU (n − 1) as input. Then, "conv+bn+relu" processing is done on the two features so that the two feature channels have the same number, where $g_{f-1}$ reuses the deconvolution with a step size of 2 to achieve consistency in the size of the feature map [45]. Finally, the corresponding positions of the features are multiplied to obtain $g_f$. The formula is defined as follows (Equation (8)):

$$v = D_{c0}(C_{3\times3}(g_{f-1})), u = C_{3\times3}(m), g_f = v \otimes u \tag{8}$$

where $C_{3\times3}$ represents the ConvBnRelu operation with a convolution kernel size of $3 \times 3$. $D_{c0}$ represents the deconvolution operation to achieve 2 times upsampling. $\otimes$ represents the matrix multiplication operation. $g_f$ is the feature map of the output of the gate unit.

The second part is the refinement unit (RU), which uses the result generated by $g_f$ and the previous layer of RU (n − 1) as input. After $g_f$ is input, the "conv+bn+relu" operation is performed to change the number of channels to 2, and then added to the RU (n − 1) result $r_{f-1}$ by concat. Finally, the output result is obtained by deconvolution. The formula is (Equation (9)):

$$w_f = C_{3\times3}(g_f), \gamma = w_f \oplus r_{f-1}, r_f = D_{c1}(\gamma) \tag{9}$$

where $C_{3\times3}$ represents the ConvBnRelu operation with a convolution kernel size of $3 \times 3$. $D_{c1}$ represents the deconvolution operation. $\oplus$ represents feature concatenation. $r_f$ is the feature map output after the refinement unit.

## 3. Results

The experiments conducted in this study use pycharm under Windows to write the program. The configuration of the experimental platform is an Intel Core i9-9900K CPU and an Nvidia RTX2080Ti (11 G) GPU. Both Intel and Nvidia are headquartered in Santa Clara, CA, USA.

### 3.1. Dataset

To verify the neural network effectiveness in extracting road extraction in different environments, the DeepGlobe Road Extraction dataset [36] is used. The original size of the

DeepGlobe Road Extraction dataset is 19,584 × 19,584, and the ground resolution of the image pixels is 0.5 m/pixel. There are various types of roads, including those in mountains, cities, and suburbs. To save memory and adapt to the network structure, the image size is cropped to 512 × 512. The selected training set contains 6152 images and corresponding label images, and the verification set contains 300 images and corresponding labels. To improve the generalization ability of the model and reduce overfitting, data-enhancement operations such as rotation and mirroring are performed on the dataset.

### 3.2. Metrics

In this study, three metrics commonly used in semantic segmentation are used to evaluate the performance of the different algorithms in the experiments: accuracy, recall [43], and F1-score [46], which are defined below. See Table 1 for the four types of road-extraction results.

**Table 1.** Four types of road extraction results. GT is the ground truth, and P is the prediction, TP is a positive sample that is correctly judged, and FP is a positive sample that is incorrectly judged. TN is a negative sample that is correctly judged and FN is a negative judgement (false judgement).

|  | **P Is 1** | **P Is 0** |
|---|---|---|
| GT is 1 | TP | FN |
| GT is 0 | FP | TN |

The three metrics in Table 1 are calculated as follows (Equations (10)–(12)).
Recall (R):

$$recall = \frac{TP}{TP + FN} \tag{10}$$

Precision (P):

$$precision = \frac{TP}{TP + FP} \tag{11}$$

F1-score (F):

$$F1 = \frac{2TP}{2TP + FN + FP} \tag{12}$$

### 3.3. Experimental Results

In this work, the proposed GMR-Net model calculates the loss value of the output four scale images and sets the loss function as the sum of the loss values of each scale. RMSprop [30] is selected as the optimizer, the parameter LR is set to $1 \times 10^{-4}$, w_decay is $1 \times 10^{-4}$, and the training batch is 6. To more intuitively observe the GMR-Net road extraction effect, we perform road extraction on the remote sensing images of the verification set. The final segmentation result is shown in Figure 4. The final extraction accuracy on the verification set is given in Table 2. The accuracy is 87.97%, the recall is 88.86%, and the F1-score is 88.41%.

**Table 2.** Results of the evaluation metrics.

|  | **Precision (%)** | **Recall (%)** | **F1-Score (%)** |
|---|---|---|---|
| Output (512) | 87.97 | 88.86 | 88.41 |

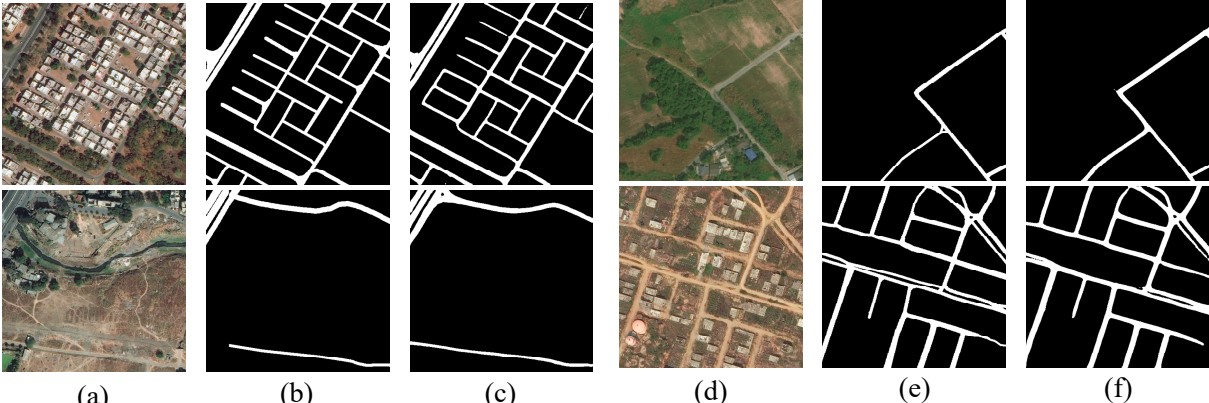

(a)　　　(b)　　　(c)　　　(d)　　　(e)　　　(f)

**Figure 4.** The results of road extractions using the proposed GMR-Net model. (**a**,**d**) Original images, (**b**,**e**) corresponding ground truths, and (**c**,**f**) predictions from the GMR-Net method.

The extraction and fusion of local and global information have a significant influence on the road extraction performance from high-resolution remote sensing images. The three modules proposed in this paper are the residual network with global attention, MDC, and GR. The first two modules extract local and global information, and the latter module fully fuses the extracted information. First, a comparative experiment is carried out on the network encoding part to verify that the migrated Resnet34 and global attention modules can effectively improve the model's performance and to validate the feasibility of the road extraction model. Figure 5 shows that during the model training, the loss value of each scale on the road verification set of the DeepGlobe satellite-image changes as the number of iterations increases. It can be seen that after the migration of the Resnet34 parameter and the addition of the global attention module, the initial loss value of the network is low, the convergence speed is fast, and the loss value is always kept to the minimum during the training process. Table 3 shows the final fitting loss value of each scale on the verification set. The final loss value of the encoding part designed in this paper is lower than other encoding parts at the 512 scales, specifically 0.11587. This proves that transfer learning and global attention modules can help the model improve the accuracy of road extraction and speed up the fitting speed in the training process.

**Table 3.** Loss values at different scales, where the values in bold are the best.

|  | ResNet34 | ResNet34 + Gc | ResNet34 + Gc + Transfer |
|---|---|---|---|
| Output (64) | 0.17155 | 0.13932 | **0.11765** |
| Output (128) | 0.16586 | 0.13912 | **0.11734** |
| Output (256) | 0.15965 | 0.13738 | **0.11626** |
| Output (512) | 0.15788 | 0.13710 | **0.11587** |
| Output_sum | 0.65494 | 0.55292 | **0.46712** |

*3.4. Ablation Experiments*

To further illustrate the performance of each of our proposed modules, we performed ablation experiments. Next, the models that do not contain the MDC and GR structures are compared to illustrate the contribution of the MDC and GR to the road extraction task (Figure 6). Table 4 shows the loss value on the validation set. The same training data set and encoding part are used. It is found that when the designed model does not include MDC and GR, the highest loss value is 0.13871. When the model includes these two modules, the loss value drops to the lowest at 0.11587. It can be seen that whether it is extracting the overall structure of the road or the details of the road, the network including the MDC and GR modules can achieve better performance. Although the network containing MDC can extract the local and global information of the road, it does not fully integrate the local and global information by GR. The network that contains GR fully integrates the information,

but the extracted local and global information is not sufficient. Therefore, the performance of these two structures is insufficient when they exist alone. By combining MDC and GR, the network can extract and integrate local and global information, and accurately extract roads from remote sensing images.

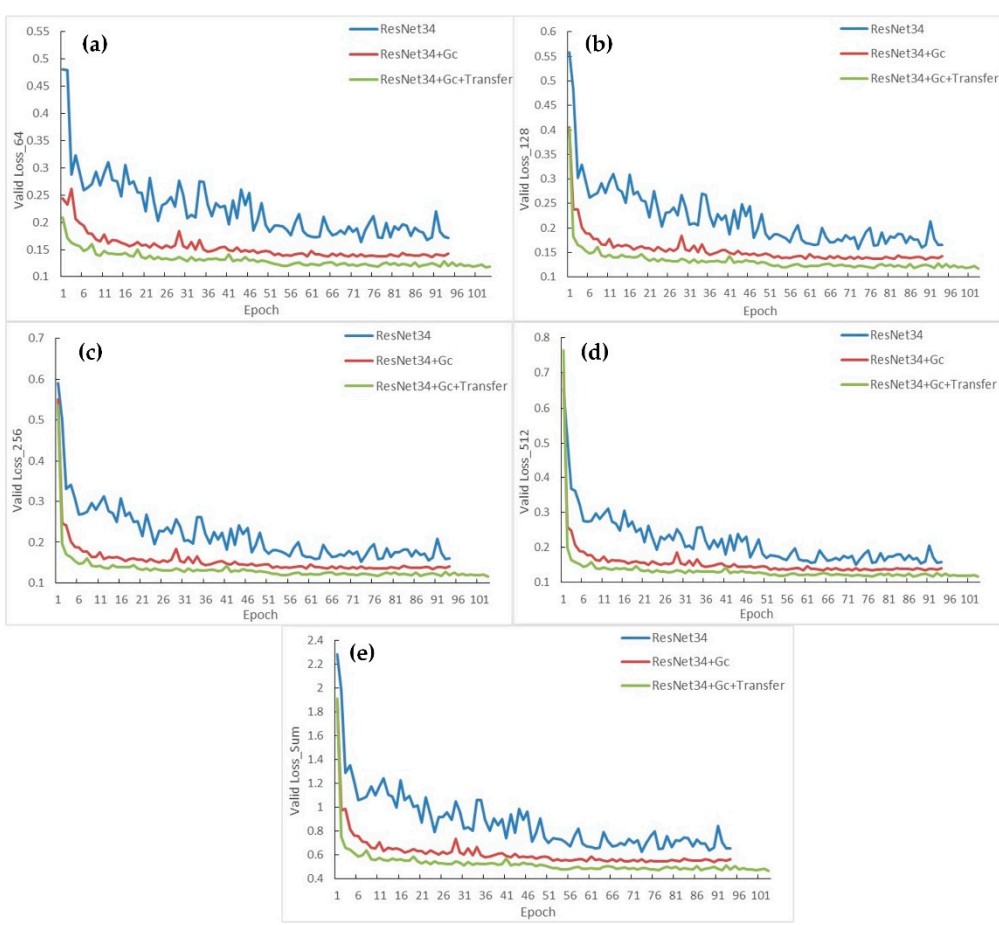

**Figure 5.** Loss value of validation from DeepGlobe dataset. Loss value of size (**a**) 64 × 64; (**b**) 128 × 128; (**c**) 256 × 256; (**d**) 512 × 512; and (**e**) sum of all scales.

**Table 4.** The results of ablation experiments. The values in bold are the best.

| Elements | Loss Value | Precision (%) | Recall (%) | F1-Score (%) |
| :---: | :---: | :---: | :---: | :---: |
| Without both | 0.13871 | 85.42 | 86.14 | 85.77 |
| Only MDC | 0.12881 | 86.58 | 87.42 | 86.99 |
| Only GR | 0.12322 | 86.34 | 87.68 | 87.00 |
| MDC and GR | **0.11587** | 87.97 | 88.86 | 88.41 |

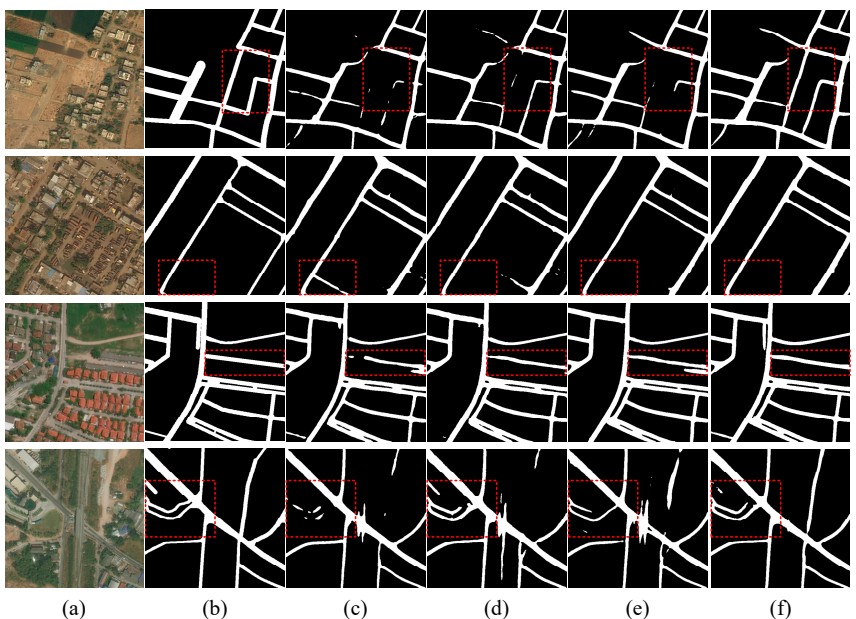

**Figure 6.** The results of road extraction from the remote sensing imagery with and without MDC and GR. (**a**) Input images; (**b**) ground truth; (**c**) prediction results without MDC and GR; (**d**) prediction results without GR; (**e**) prediction results without MDC; (**f**) prediction results using the GMR-Net model. The red-dashed frame indicates the local details of semantic segmentation.

## 4. Discussion

### 4.1. Analysis of the GMR-Net Road Extraction Results in Different Environments

To evaluate the relative performance of GMR-Net on the DeepGlobe Road Extraction dataset, three other methods (U-Net, PSPNet, RSANet, and D-Linknet) are used to extract different types of road images. The results are stitched together to obtain a remote-sensing image with a 1024 × 1024 resolution. All methods are trained and tested on the same datasets.

In research areas A and B, the roads are located in forest and desert areas. Road segmentation is mainly affected by vegetation coverage and tree shadows, which can easily lead to omissions. To evaluate the model's effectiveness in this environment, some common methods are compared, and the results are shown in Table 5. It can be seen that the various methods are generally highest in Recall and lowest in Precision, which can prove that misjudgments are potentially high in such environments. In addition, the GMR-Net model has the highest Precision and F1-score, with an average increase of 5% and 4%, respectively, compared to D-Linknet. From Figure 7, it is found that the D-Linknet and RSANet can complete the extraction of most of the blocked roads (red-dashed frame) and avoids the edge aliasing, but it cannot be well divided for the areas with more serious blocking. PSPNet performs the worst under this condition. Considering that when roads are covered by vegetation, shadows, etc., local information is easily lost. Therefore, the proposed GMR-Net pays attention to more detailed local information through dilated convolution of different receptive fields and uses GR to enhance the semantics of the local information. In the end, the GMR-Net can better extract the covered road and ensure road continuity as much as possible.

**Table 5.** A results comparison between the GMR-Net method and other methods in research areas A and B. The values shown in bold are the best.

| | Area A | | | Area B | | |
|---|---|---|---|---|---|---|
| | **P (%)** | **R (%)** | **F (%)** | **P (%)** | **R (%)** | **F (%)** |
| U-Net | 81.42 | **99.64** | 88.36 | 72.36 | 98.73 | 80.74 |
| PSPNet | 79.62 | 99.39 | 87.03 | 59.69 | **99.71** | 68.33 |
| RSANet | 79.76 | 99.06 | 87.92 | 75.68 | 99.52 | 84.26 |
| D-Linknet | 83.19 | 98.41 | 89.29 | 77.53 | 99.57 | 87.18 |
| GMR-Net | **89.72** | 98.99 | **93.83** | **83.71** | 99.17 | **90.26** |

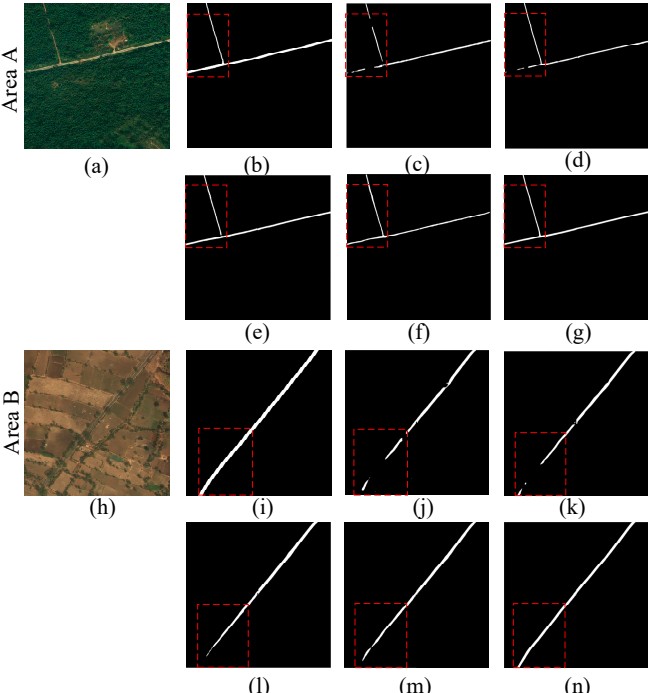

**Figure 7.** A comparison of classification results in study areas A and B. (**a**,**h**) Input images; (**b**,**i**) ground truth; (**c**,**j**) U-Net; (**d**,**k**) PSPNet; (**e**,**l**) RSANet; (**f**,**m**) D-Linknet; and (**g**,**n**) GMR-Net. The position of the red-dashed frame indicates the area where the judgment is missing.

Research areas C and D belong to the suburbs, and the surrounding environment mainly comprises wasteland and housing construction. Roads are usually located in barren land, and the spectral information is relatively close to the land, which can easily result in false judgment. The comparison results of road extraction in this type of area are shown in Table 6 below. It is found that Recall is generally low, indicating that this type of area is indeed prone to misjudgment. The method comparison shows that GMR-Net has the highest Recall score, and its F1-score is higher than that of RSANet (U-Net) by approximately 5% (9%). It can be seen from the yellow-dashed frame area in Figure 8 that there are some traces similar to the color and structure of the road in the wasteland. If only the local information of the yellow-dashed frame is considered and the focus on global information is not strengthened, it is easily mistaken as a road. As shown by U-Net and PSPNet, these networks misclassify the empty area. The area that does not contain roads in the original label is divided into roads, resulting in over-segmentation. Although RSANet's performance has been improved, there are still several errors for several small traces. The same is true for D-LinkNet. The GC-block referenced by GMR-Net extracts global information on a lightweight basis, focusing on the channel-to-channel dependency. MDC also pays attention to both global and local information. The method advanced in

this study avoids the interference of this kind of environment, distinguishes the road from the wasteland, and reduces false judgment of such situations.

**Table 6.** A results comparison between the GMR-Net method and other methods in research areas C and D, where the values shown in bold are the best.

| | Area C | | | Area D | | |
|---|---|---|---|---|---|---|
| | **P (%)** | **R (%)** | **F (%)** | **P (%)** | **R (%)** | **F (%)** |
| U-Net | 88.06 | 61.50 | 72.43 | 71.21 | 70.61 | 70.91 |
| PSPNet | 85.21 | 63.25 | 72.45 | 65.61 | 74.06 | 69.42 |
| RSANet | 88.69 | 62.45 | 73.29 | **77.56** | 68.83 | 72.94 |
| D-Linknet | **89.56** | 64.98 | 75.10 | 74.10 | 71.64 | 72.85 |
| GMR-Net | 89.16 | **77.56** | **82.69** | 72.38 | **80.34** | **75.73** |

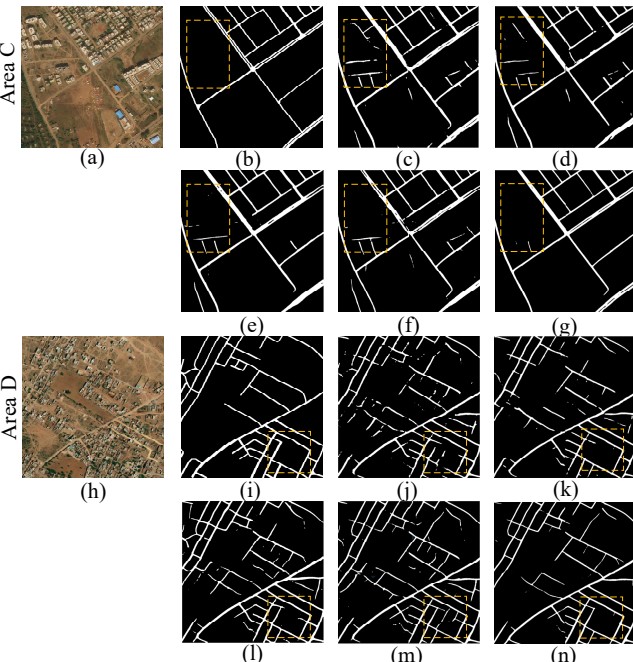

**Figure 8.** A classification results comparison of study areas C and D. (**a**,**h**) Input images; (**b**,**i**) ground truth; (**c**,**j**) U-Net; (**d**,**k**) PSPNet; (**e**,**l**) RSANet; (**f**,**m**) D-Linknet; and (**g**,**n**) GMR-Net. The position of the yellow-dashed frame indicates areas where false judgment occurs.

Research areas E and F belong to urban roads. Vehicles on highways, vegetation, city buildings and cement structures similar to road textures all interfere with road extraction. Table 7 shows that the F1-score of GMR-Net is nearly 4% higher than that of D-Linknet, and is much higher than those of U-Net and PSPNet. Precision increased by nearly 5%. The segmentation diagram in Figure 9 demonstrates that U-Net and PSPNet are more seriously affected by noise and perform poorly in urban road segmentation. These methods cannot accurately divide an urban road under the condition of occlusion and mistake the gaps between buildings as roads. GMR-Net benefits from the increased attention to local and global information and has achieved relatively satisfactory performance in urban environments, segmenting more continuous roads.

**Table 7.** A results comparison between the GMR-Net method and other methods in research areas E and F, where the values shown in bold are the best.

| | Area C | | | Area D | | |
|---|---|---|---|---|---|---|
| | **P (%)** | **R (%)** | **F (%)** | **P (%)** | **R (%)** | **F (%)** |
| U-Net | 74.26 | 85.38 | 79.42 | 77.82 | 80.03 | 78.73 |
| PSPNet | 67.83 | 86.92 | 76.14 | 71.80 | 83.72 | 77.17 |
| RSANet | 80.15 | 87.03 | 83.22 | **79.27** | 82.08 | 80.39 |
| D-Linknet | 78.83 | **90.24** | 84.15 | 72.63 | 88.01 | 79.45 |
| GMR-Net | **85.02** | 90.04 | **87.43** | 77.01 | **90.74** | **83.17** |

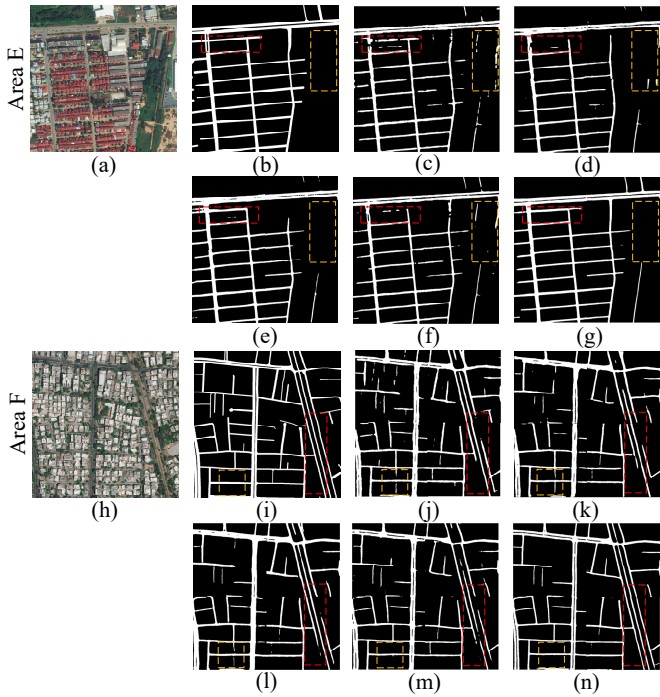

**Figure 9.** A classification results comparison for study areas E and F. (**a**,**h**) Input images; (**b**,**i**) ground truth; (**c**,**j**) U-Net; (**d**,**k**) PSPNet; (**e**,**l**) RSANet; (**f**,**m**) D-Linknet; and (**g**,**n**) GMR-Net. The red-dashed frame shows areas where judgement is missing and the yellow-dashed frame is the false judgment area.

### 4.2. Model Generalization Ability Analysis

Based on the above analysis, it is found that the GMR-Net model achieves a high segmentation accuracy on the DeepGlobe Road Extraction dataset and effectively eliminates false judgment and omission.

To further verify the generalization ability of GMR-Net, the model is trained and tested on the Massachusetts Roads dataset [37], which covers more than 2600 km² of remote-sensing satellite images of Massachusetts, with a size of 1500 × 1500 and a ground resolution of 1 m/pixel. The Massachusetts Roads dataset originally consisted of 1171 images. After cropping, 3384 training sets and 156 validation sets are obtained, with a resolution of 512 × 512.

In Figure 10, the final segmentation comparison results show that the GMR-Net model can extract a more complete road, and is closest to the label. It is also found that U-Net, PSPNet, RSANet, and D-LinkNet have omission areas (red-dashed frame) due to the complex and curved nature of the road network in this area. In particular, from the second row of the figure, the road network in the red-dashed frame is crisscrossed, and the distance between the roads is small. Except for the GMR-Net model, all other methods have discontinuous and incomplete results. The yellow-dashed frame marks

the false judgment of the road network. U-Net and other networks recognize the ground features similar to the road texture and color characteristics as roads, leading to excessive segmentation. The GMR-Net algorithm effectively avoids this phenomenon. Table 8 shows the evaluation indicators of each model on the two verification datasets. The Precision, Recall, and F1-score of GMR-Net on the DeepGlobe Road Extraction dataset reach 87.97, 88.86, and 87.38%, respectively. On the Massachusetts Roads dataset, these values are 83.91, 87.60, and 85.70%, respectively. From the time-complexity comparison, this method takes a long time since the model is not computationally lightweight enough. Overall, although the extraction speed of this method is slow, it can obtain higher accuracy and has a certain generalization ability.

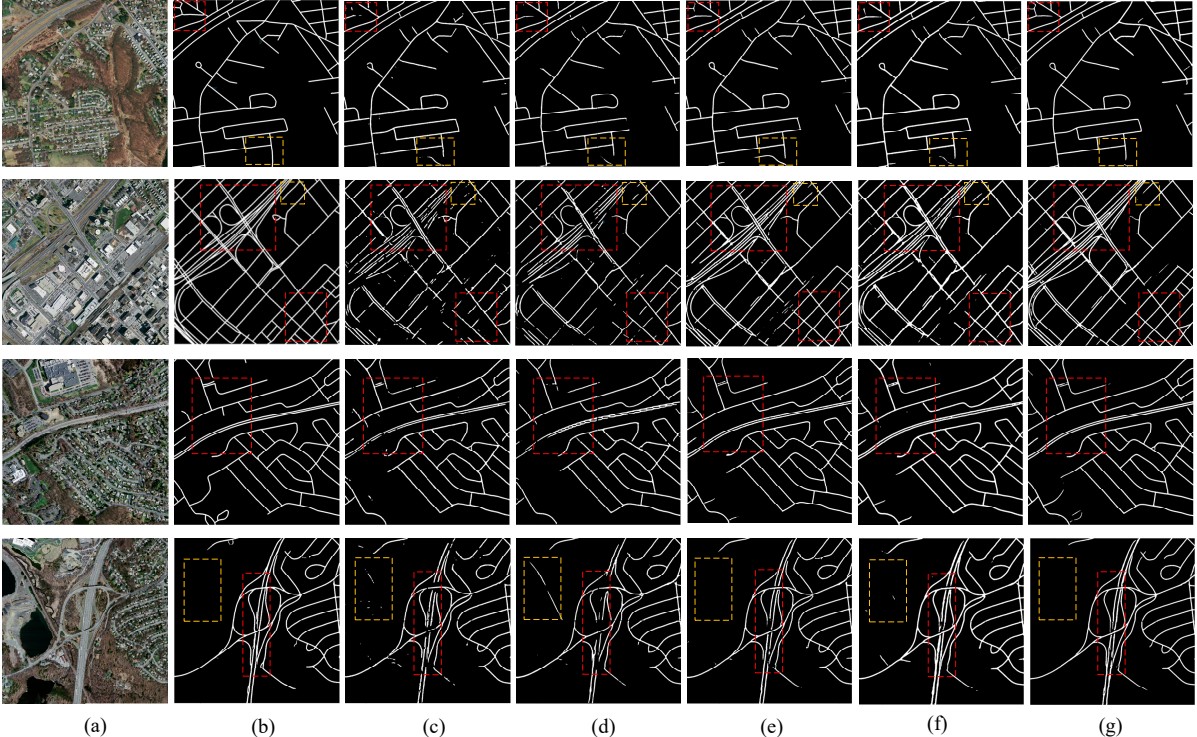

(a)　　　(b)　　　(c)　　　(d)　　　(e)　　　(f)　　　(g)

**Figure 10.** A comparison of classification results on the Massachusetts Roads dataset. (**a**) Input images; (**b**) ground truth; (**c**) U-Net; (**d**) PSPNet; (**e**) RSANet; (**f**) D-Linknet; and (**g**) GMR-Net. The red-dashed frame shows the area where the judgement is missing and the yellow-dashed frame is the false judgment area.

**Table 8.** Using DeepGlobe Road-Extraction and Massachusetts Road datasets, the results of the GMR-Net method are compared with other methods, in which the values shown in bold are best.

|  |  | U-Net | PSPNet | RSANet | D-LinkNe | GMR-Net |
|---|---|---|---|---|---|---|
| DeepGlobe Road Extraction dataset | P(%) | 78.50 | 83.73 | 80.19 | 82.01 | **87.97** |
|  | R(%) | 83.19 | 73.44 | 83.16 | 87.40 | **88.86** |
|  | F(%) | 78.76 | 76.67 | 80.34 | 84.62 | **87.38** |
|  | T(ms) | **24** | 40 | 41 | 56 | 60 |
| Massachusetts Roads dataset | P(%) | 70.24 | 65.23 | 80.57 | **85.80** | 83.91 |
|  | R(%) | 82.90 | 81.74 | 85.78 | 81.97 | **87.60** |
|  | F(%) | 75.56 | 72.10 | 82.95 | 83.84 | **85.70** |
|  | T(ms) | **24** | 40 | 41 | 56 | 60 |

*4.3. Feature Visual Analysis*

To explore the process of extracting road features from the network, the feature visualization analysis of the encoding part is performed (Figure 11).

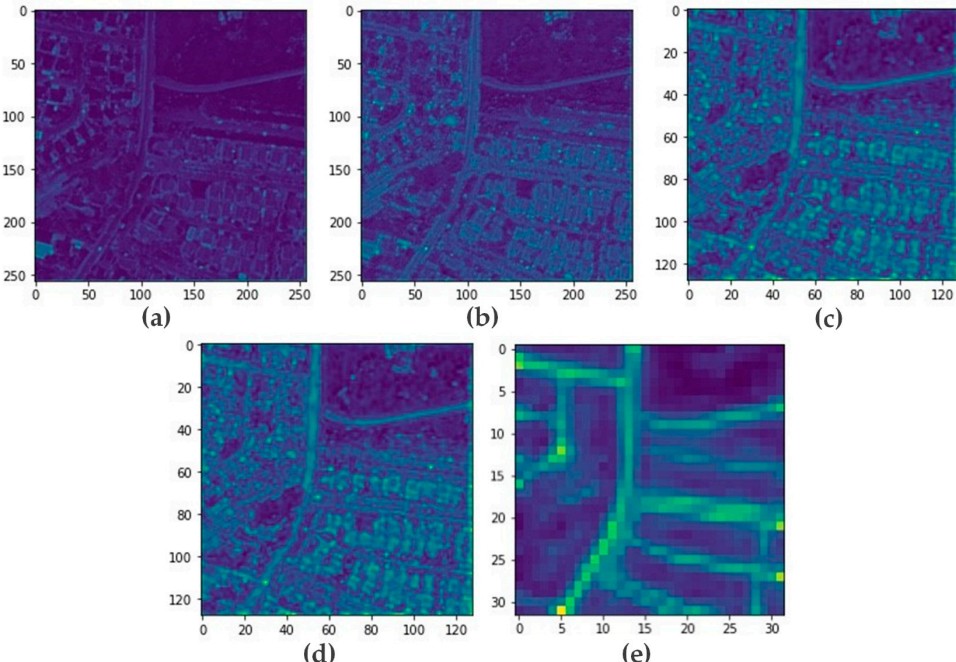

**Figure 11.** The visualization of features at each stage. (**a**) First stage visualization; (**b**) second stage visualization; (**c**) third stage visualization; (**d**) fourth stage visualization; (**e**) fifth stage visualization.

Using a randomly selected $512 \times 512 \times 3$ high-resolution remote sensing image as input, the image passes through a convolutional layer with a convolution kernel size of 7 and a step size of 2 in the first stage of the encoding process. The size of the output image becomes $256 \times 256 \times 64$ (Figure 11a). The remote sensing image is only convolved once, and the extracted features are weak in semantics, but the resolution is higher and the detailed features are more.

In the second stage, the feature image passes through the 3 residual modules and the global attention module, resulting in an image size of $256 \times 256 \times 64$ (Figure 11b). It can be seen that after the first global residual block, the network has stronger semantics than the features extracted in the first stage.

In the third stage, the feature image passes through the 4 residual modules and the global attention module, and the output image size becomes $128 \times 128 \times 128$ (Figure 11c). At this time, the network has passed the second global residual block, the feature channel of the image is increased, the resolution is reduced, and the feature semantics is further enhanced.

In the fourth stage, the feature image passes through 6 residual modules, and the output image size becomes $64 \times 64 \times 256$ (Figure 11d). The feature channel increases and the resolution is further reduced.

In the fifth stage, the feature image passes through 3 residual modules, and the output image size becomes $32 \times 32 \times 512$ (Figure 11e). At this stage, the feature resolution extracted by the network is reduced to the lowest level, and more detailed information is lost. However, it extracts features with strong semantics. The visual feature map demonstrates that the network focuses on the roads and avoids interference from buildings.

To further illustrate the effect of feature extraction in the GMR-Net coding part, the visualization of the deep partial feature channel is provided in Figure 12. It can be observed that GMR-Net focuses attention on the target road, avoiding interference from background features. Compared with D-Linknet, more complete and continuous road features are extracted, and the road structure is clearer. The effectiveness of the global information for extracting road features is confirmed, and the extracted road features are more continuous.

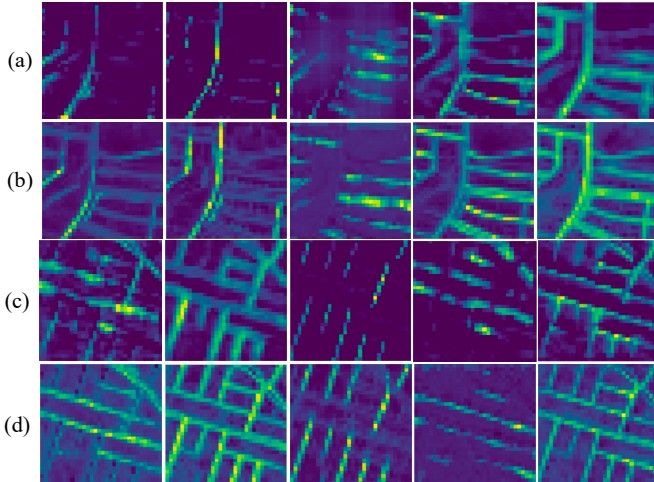

**Figure 12.** A visualization of different channels' features. Samples (**a**,**b**) are features extracted by the encoding part in D-Linknet. Samples (**c**,**d**) are extracted features after introducing GC-block.

## 5. Conclusions

Roads have the characteristics of narrowness, complexity and connectivity, posing different problems to road extraction in different environments. It is particularly important to focus on the information in global and local contexts and remove interference from other features. Therefore, a neural network for remote-sensing road extraction based on the fusion of local and global information is proposed in this study. The proposed network, GMR-Net, consists of three parts. The first part is the GC-resnet. A GCblock, which extracts deep features, is used to realize global context modeling and capture the relationship between channels. The second part is the MDC, through which the context information of different regions can be aggregated. The third part efficiently combines the global and local information through GR, filters the ambiguous features in the encoding stage, and gradually refines the segmentation details. This model is separately tested, compared and analyzed using the DeepGlobe Roads and Massachusetts Roads datasets, and the extraction results of U-Net, PSPNet, RSANet, and D-Linknet. It is found that the GMR-Net can effectively extract road features, ensure the continuity and integrity of road extraction, and show good generalization ability. Although the proposed method improves the accuracy of road segmentation, the speed of road extraction is slow and the network is not computationally lightweight enough. Therefore, there is still room for improvement in the proposed method. Ensuring high-precision road extraction while accelerating the segmentation speed is worth further investigation.

Transformer structures have excellent global information modeling capability and are currently advanced and competitive in the field of computer vision. Our future work will design a Transformer-based model for road extraction tasks and investigate the potential of Transformer structures for road extraction in remote sensing images.

**Author Contributions:** All authors contributed significantly to this manuscript. Conceptualization, Z.Z. and X.S.; methodology, Z.Z.; software, Y.L.; validation, Z.Z., X.S. and Y.L.; formal analysis, X.S.; data curation, Z.Z. and X.S.; writing—original draft preparation, Z.Z.; writing—review and editing, Z.Z. and X.S.; visualization, Y.L.; supervision, Z.Z.; project administration, X.S.; funding acquisition, X.S. and Z.Z. All authors have read and agreed to the published version of the manuscript.

**Funding:** This research was funded by National Innovation and Entrepreneurship Training Program of China for College Students, (Grant No. 202210055022), and the National Natural Science Foundation of China (Grant No. 72074127).

**Institutional Review Board Statement:** Not applicable.

**Informed Consent Statement:** Not applicable.

**Data Availability Statement:** Not applicable.

**Acknowledgments:** We thank National Natural Science Foundation of China and National Key Research and Development Program of China for the research support.

**Conflicts of Interest:** The authors declare no conflict of interest.

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
