# Peer review of "GMR-Net: Road-Extraction Network Based on Fusion of Local and Global Information"

_remotesensing, doi:10.3390/rs14215476_

Round 1

Reviewer 1 Report

The paper proposed a method GMR-Net for road network extraction from remote sensing images. Local and global information was combined in the data treatment process, and promising results were obtained.  Some detailed suggestions are listed as follows.

1. Please reorganize the introduction about related works. There is no referecnes in recent 2 years. Most references have little relationship about road extraction. Specially, there is few remote sensing-related citations. Please carefully reorganize section Introduction, remove all the unrelated citations, and provide more investigation about most recent works about road extraction from remote sensing images.

2. It would be better to futher polish the language expression. The content was not easy to undersand in some parts of the paper. 

3. More explaination about the parameters of the functions adopted might help to strengthen the theoretical basis of the method.

Author Response

Dear Reviewer :

The paper proposed a method GMR-Net for road network extraction from remote sensing images. Local and global information was combined in the data treatment process, and promising results were obtained. Some detailed suggestions are listed as follows.

Reply to review 1:

Firstly, we appreciate that you gave us a chance of revision to improve our manuscript to a level suitable for publication in Remote Sensing. We also want to express our deep thanks for your positive comments. The comments are replied as follows:

Question 1: Please reorganize the introduction about related works. There is no referecnes in recent 2 years. Most references have little relationship about road extraction. Specially, there is few remote sensing-related citations. Please carefully reorganize section Introduction, remove all the unrelated citations, and provide more investigation about most recent works about road extraction from remote sensing images.

Response to Reviewer 1 comment No.1:

Thank you very much for your valuable suggestions. We have removed some old citations and added some new references from the last two years on road extraction work. As indicated in the section of

References, a large number of documents have been updated

Question 2: It would be better to further polish the language expression. The content was not easy to understand in some parts of the paper.

Response to Reviewer 1 comment No.2:

Thank you very much for your valuable suggestions. We have rechecked the full paper and reorganized some linguistic expressions.

Question 3: More explaination about the parameters of the functions adopted might help to strengthen the theoretical basis of the method.

Response to Reviewer 1 comment No.3:

Thank you very much for your valuable suggestions. Based on the reviewers' comments, we have reorganized and revised the functions in the paper. Also some detailed parameter explanations have been added, as shown in line 185, 228, 235, 263, 305 and 314.

Reviewer 2 Report

remotesensing-1965833-peer-review-v1

This manuscript proposed a new image segmentation method for road extraction. Generally, the method is novel and the article is well written. Several points should be considered to further improve the readability.

1.       The terms and sentences should be carefully checked and account for their consistency. Here I list a few examples but there are many similar expressions in the whole manuscript that need to be revised.

a.        L44-45, what are road nodes and set nodes in the referred paper?

b.       L102-110, the contribution involves too many technical details, which were not easily understandable in the introduction part. Please brief this paragraph and move the technical details into the method part.

c.       L121-127, this part did not mention using multi-scale, so ‘each scale’ is rather strange in the sentence.

d.       L129-135, how does ‘multi-path’ related to ‘multi-scale’ is unclear. The first part is decoding, I suppose the third part is decoding using the GR? Revise the expressions and enhance the consistency.

e.       L141, why using ResNet34+GCblock should be explained.

f.        Subsection 4.3, the six stages listed here but cannot find them in the method part.

2.       Some expressions are not clear and grammar errors are included. A few examples are listed below, please check the whole manuscript.

a.       The first sentence in the abstract includes grammar error.

b.       L38-L41, the first point is not entirely true, the second point should be a sentence rather than words. Please revise this part.

c.       L47, ‘enter the initial seed point’ is not clear.

d.       L87, ‘is’ should revise as ‘are’.

e.       L108-110, this sentence does not make sense.

f.        Subsection 3.3, some numbers should be written as superscripts.

3.       All the symbols in equations should be carefully checked. Currently there are many symbols without explanation, such as αj, v1, v2, k in (1) and (2), σ in (5). The explanation of w and b is not right. Whey C is in a carved format? What do and mean?

4.       L289, the training batch is 6 is rather small, the ResNet34 in Figure 5 does not seem to converge.

5.       There lacks a discussion on the method. The current discussion is actually an analysis part. Please add a paragraph to discuss the advantages of this method over similar methods theoretically, as well as the limitations of this method.

Author Response

Dear Reviewer :

Please see the attached materials, it is the modified content. Thank you very much.

Reviewer 3 Report

(1)The numbers of training, validation, and testing subset are inconsistent with the public datasets on the DeepGlobe and Massachusetts. The author should clearly describe the way and basis for data division.

(2)There are too many outdated references for this deep learning-based road extraction method. Ref. 16 is the same as Ref.34, please carefully check the reference.

(3)The following references can be cited. Cascaded Residual Attention Enhanced Road Extraction from Remote Sensing Images which proposed cascaded attention enhancement modules considering the multi-scale spatial details of the roads. MAP-Net: Multiple attending path neural network for building footprint extraction from remote sensed imagery which efficiently extracts and fuses global and local context information through an attention-enhanced multi-path network.

(4)Its encouraged to open the source code which helps the development of the remote sense community.

Author Response

Dear Review:

(1) The numbers of training, validation, and testing subset are inconsistent with the public datasets on the DeepGlobe and Massachusetts. The author should clearly describe the way and basis for data division.

(2) There are too many outdated references for this deep learning-based road extraction method. Ref. 16 is the same as Ref.34, please carefully check the reference.

(3) The following references can be cited. ‘Cascaded Residual Attention Enhanced Road Extraction from Remote Sensing Images’ which proposed cascaded attention enhancement modules considering the multi-scale spatial details of the roads. ‘MAP-Net: Multiple attending path neural network for building footprint extraction from remote sensed imagery’ which efficiently extracts and fuses global and local context information through an attention-enhanced multi-path network.

(4) It’s encouraged to open the source code which helps the development of the remote sense community.

Reply to Review 3:

Firstly, we appreciate that you gave us a chance of revision to improve our manuscript to a level suitable for publication in Remote Sensing. We also want to express our deep thanks for your positive comments. The comments are replied as follows:

Question 1: The numbers of training, validation, and testing subset are inconsistent with the public datasets on the DeepGlobe and Massachusetts. The author should clearly describe the way and basis for data division.

Response to Reviewer 3 comment No.1:

Thank you very much for your valuable suggestions. The original size of DeepGlobe Road Extraction dataset is 19584×19584. Due to the limitation of our working platform, we selected some images from the original dataset and cropped them to 512×512 size. We selected 6152 images as the training set and 300 images as the validation set. For the Massachusetts Roads dataset, we did the same operation of cropping the original image size to 512×512. So the training set used in the paper is not consistent with the division of the training and validation sets in the original dataset.

Question 2: There are too many outdated references for this deep learning-based road extraction method. Ref. 16 is the same as Ref.34, please carefully check the reference.

Response to Reviewer 3 comment No.2:

Thank you very much for your valuable suggestions. We have deleted Reference 16 and replaced some references in the manuscript with the latest.

Question 3: The following references can be cited. ‘Cascaded Residual Attention Enhanced Road Extraction from Remote Sensing Images’ which proposed cascaded attention enhancement modules considering the multi-scale spatial details of the roads. ‘MAP-Net: Multiple attending path neural network for building footprint extraction from remote sensed imagery’ which efficiently extracts and fuses global and local context information through an attention-enhanced multi-path network.

Response to Reviewer 3 comment No.3:

Thank you very much for your valuable suggestions. We added two articles suggested by reviewers to the manuscript.

Question 4: It’s encouraged to open the source code which helps the development of the remote sense community.

Response to Reviewer 3 comment No.4:

Thank you very much for your valuable suggestions. Readers can contact the email address (sunxuan@mail.nankai.edu.cn) of the corresponding author to get the code of the model. At the same time, after the paper is officially published, we will open source on Github.

Reviewer 4 Report

This paper proposed a Road-Extraction Network Based on Fusion of Local and Global Information. Overall, the structure of this paper is well organized, and the presentation is clear. However, there are still some crucial problems that need to be carefully addressed before a possible publication. More specifically,

1.       A deep literature reviews should be given, particularly advanced and SOTA deep learning or AI methods. Therefore, the reviewer suggests discussing some related works by analyzing the following papers in the revised manuscript, e.g.,  10.1109/TGRS.2020.3015157

2.       Please clarify the contributions to this field, for example, which are the existing ones and which are your own ones?

3.       What are the differences in techniques between the proposed method and existing methods?

4.       The ablation analysis should be given to show the performance gain by using different strategies and techniques.

5.       Some future directions should be pointed out in the conclusion.

Author Response

Dear Review:

This paper proposed a Road-Extraction Network Based on Fusion of Local and Global Information. Overall, the structure of this paper is well organized, and the presentation is clear. However, there are still some crucial problems that need to be carefully addressed before a possible publication.

Reply to Review 4:

Firstly, we appreciate that you gave us a chance of revision to improve our manuscript to a level suitable for publication in Remote Sensing. We also want to express our deep thanks for your positive comments. The comments are replied as follows:

Question 1: A deep literature reviews should be given, particularly advanced and SOTA deep learning or AI methods. Therefore, the reviewer suggests discussing some related works by analyzing the following papers in the revised manuscript, e.g., 10.1109/TGRS.2020.3016820, 10.1109/TGRS.2020.3015157, 10.1109/TCYB.2022.3175771.

Response to Reviewer 4 comment No.1:

Thank you very much for your valuable suggestions. We discussed the relevant work according to the paper you suggested.

Question 2: Please clarify the contributions to this field, for example, which are the existing ones and which are your own ones?

Response to Reviewer 4 comment No.2:

Thank you very much for your valuable suggestions. Attention mechanism is a hotspot in current research. How to learn useful features from images. The attention residual module we proposed can better learn the features related to the road. The dilation convolution increases the receptive field of the image without increasing the amount of computation. In the proposed method, we designed a multi-path expansion convolution, which can further learn the road related features of different receptive fields. The learning of this information is useful for the accurate extraction of roads and the suppression of background noise.

Question 3: What are the differences in techniques between the proposed method and existing methods?

Response to Reviewer 4 comment No.3:

Thank you very much for your valuable suggestions. Different from other methods, in this paper, we propose a residual module with attention to learn the global and other aggregated information of the image. Multi path expansion convolution is used to extract road features of different scales. In terms of extracting remote sensing image information, our method obtains more abundant information and learns more semantic information.

Question 4: The ablation analysis should be given to show the performance gain by using different strategies and techniques.

Response to Reviewer 4 comment No.4:

Thank you very much for your valuable suggestions. Based on the original structure, we have added the ablation experiment, as shown in line 384.

Question 5: Some future directions should be pointed out in the conclusion.

Response to Reviewer 4 comment No.5:

Thank you very much for your valuable suggestions. In the final summary, we added research prospects, which will guide our next work direction, as shown in line 584.

Round 2

Reviewer 1 Report

I suggest accept the current version.

Reviewer 2 Report

Previous comments are well responsed.

Reviewer 4 Report

The authors have well addressed the reviewer's concerns. No more comments.